# De Novo Development of Mitochondria-Targeted Molecular Probes Targeting Pink1

**DOI:** 10.3390/ijms23116076

**Published:** 2022-05-28

**Authors:** Shulamit Fluss Ben-Uliel, Faten Habrat Zoabi, Moriya Slavin, Hadas Sibony-Benyamini, Nir Kalisman, Nir Qvit

**Affiliations:** 1The Azrieli Faculty of Medicine in the Galilee, Bar-Ilan University, Safed 1311502, Israel; shulamit.ben-uliel@biu.ac.il (S.F.B.-U.); faten.habrat1997@gmail.com (F.H.Z.); hadas.sibony-benyamini@biu.ac.il (H.S.-B.); 2Institute of Life Sciences, The Hebrew University of Jerusalem, Jerusalem 9190401, Israel; moriya.slavin@mail.huji.ac.il (M.S.); nirka@mail.huji.ac.il (N.K.)

**Keywords:** Pink1, mitophagy, bioactive peptides, peptidomimetics, backbone cyclization, protein-protein interactions, protein-peptide interactions, therapeutic peptides, mitochondria, molecular probes

## Abstract

Mitochondria play central roles in maintaining cellular metabolic homeostasis, cell survival and cell death, and generate most of the cell’s energy. Mitochondria maintain their homeostasis by dynamic (fission and fusion) and quality control mechanisms, including mitophagy, the removal of damaged mitochondria that is mediated mainly by the Pink1/Parkin pathway. Pink1 is a serine/threonine kinase which regulates mitochondrial function, hitherto many molecular mechanisms underlying Pink1 activity in mitochondrial homeostasis and cell fate remain unknown. Peptides are vital biological mediators that demonstrate remarkable potency, selectivity, and low toxicity, yet they have two major limitations, low oral bioavailability and poor stability. Herein, we rationally designed a linear peptide that targets Pink1 and, using straightforward chemistry, we developed molecular probes with drug-like properties to further characterize Pink1. Initially, we conjugated a cell-penetrating peptide and a cross-linker to map Pink1’s 3D structure and its interaction sites. Next, we conjugated a fluorescent dye for cell-imaging. Finally, we developed cyclic peptides with improved stability and binding affinity. Overall, we present a facile approach to converting a non-permeable linear peptide into a research tool possessing important properties for therapeutics. This is a general approach using straightforward chemistry that can be tailored for various applications by numerous laboratories.

## 1. Introduction

Mitochondria play central roles in maintaining cellular metabolic homeostasis, cell survival, and cell death. They are also the so-called “powerhouses” of cells as they produce most of the cellular adenosine triphosphate (ATP). Mitochondrial homeostasis is maintained by a series of protective mechanisms, such as fission, fusion, and mitophagy, the degradation and removal of selectively damaged or dysfunctional mitochondria via autophagy. Mitophagy is mediated mainly by the Pink1 (Phosphatase-and-tensin-homolog-instigated putative kinase 1)/Parkin pathway [1].

Pink1 (aka BRPK and PARK6) is a 63 kDa mitochondrial serine/threonine-protein kinase encoded by the *PINK1* gene that was first identified in 2001 as a target of phosphatase and tensin homolog (PTEN) in cancer cells [2]. Although Pink1 was initially identified as a gene upregulated in cancer cells, the specific role of Pink1 in tumorgenesis has not been fully understood yet. Interestingly, the role of Pink1 has been widely studied in Parkinson’s disease since mutations in the *PINK1* gene were identified in 2004 as a cause of early-onset Parkinson’s disease [3]. Parkinson’s disease (PD) is a long-term degenerative disorder of the central nervous system that mainly affects the motor system. It is a progressive disorder that is caused by the degeneration of nerve cells in the part of the brain called the substantia nigra, which controls movement. Currently, there is no cure for Parkinson’s disease. However, treatments are available to help reduce the main symptoms and maintain quality of life. One recent unconventional approach that is becoming more and more common is based on nutraceuticals, substances that are food or part of food that provide medical or health benefits. Several nutraceuticals from natural sources have been substantiated to provide neuroprotection in experimental models through multiple mechanistic pathways, i.e., mitochondrial dysfunction, neuroinflammation, oxidative stress, and protein misfolding. The intake of nutraceuticals or nutritional modifications is generally safe and can be combined with current common drug therapies to improve the patient’s quality of life and/or mitigate PD symptoms [4,5,6].

Various studies have demonstrated numerous functions of Pink1, including regulating complex I activity and maintaining neuronal viability in response to stress, yet regulation of mitochondrial functions is considered to be one of the most critical ones. For example, it has been suggested that by regulating the enzymatic activity of complex I, Pink1 modulates the overall electron transport chain (ETC) capacity and, ultimately, the overall output levels of ATP [7]. In healthy mitochondria, under physiological conditions, mitochondria have an optimal, relatively high mitochondrial membrane potential (ΔΨm), and under these conditions, Pink1 is continuously imported into the mitochondria, and the transmembrane segment of Pink1 is cleaved and degraded. However, under unhealthy conditions (e.g., oxidative stress), when mitochondria are depolarized due to damage, the degradation of Pink1 is reduced, thus promoting Pink1 accumulation on the outer mitochondrial membrane (OMM). Accumulation of Pink1 on the impaired mitochondria recruits Parkin, which induces the degradation of the damaged mitochondria via a selective form of autophagy, named mitophagy [8,9].

The current prevailing model by which Pink1 and Parkin promote mitophagy posits that upon mitochondrial damage, Pink1 import is blocked, and it accumulates on the OMM of depolarized mitochondria. Pink1 then undergoes autophosphorylation and phosphorylates ubiquitin, which stimulates the recruitment of Parkin to the mitochondrial surface. Next, Parkin is phosphorylated by Pink1, and these events stimulate the ubiquitin-ligase activity of Parkin, allowing it to ubiquitinate further OMM targets, resulting in an additional substrate for Pink1, promoting more Parkin recruitment that leads to the recruitment of ubiquitin adaptor proteins, which in turn promote engulfment of the depolarized mitochondria by autophagosomes. This pathway is governed by various post-transcriptional modifications such as phosphorylation and ubiquitination, mediated by Pink1 and Parkin, respectively [10]. Although some aspects of Pink1 mitochondrial import and processing have been elucidated, several issues are still unclear. Herein, to better understand the mechanism by which Pink1 regulates mitochondrial morphology, we developed a linear peptide that targets Pink1, and based on this bioactive peptide, we engineered peptide-based targeted peptidomimetics (modified peptides), biomolecular probes, with various properties. These biomolecular probes are optimized for enhanced cell permeability, chemical crosslinking, and molecular imaging. Furthermore, these probes demonstrated improved bioactivity and superior stability. We suggest these biomolecular probes as novel and unique tools to further study the role of Pink1 under physiological and pathological conditions. In addition, we present a step-by-step guideline that can be used by most researchers and laboratories for the development of biomolecular probes based on identified naturally occurring and/or linear peptides for basic research studies as well as for therapeutic applications.

## 2. Results

### 2.1. Design of a Selective Pink1 Biomolecular Probe

Based on a rational design approach, we developed a selective Pink1 probe. We used an algorithm that identified similar regions in interacting proteins that are otherwise non-related and which was used to generate many effective and selective regulators of protein-protein interactions (PPIs) [11,12,13,14,15,16,17,18,19,20,21]. Optic atrophy 1 (Opa1) is a nuclear-encoded mitochondrial protein causing autosomal dominant optic atrophy, and it is a key player in mitochondrial fusion and cristae morphology regulation. Several studies have suggested that Pink1 interacts with Opa1, which may present an alternative pathway that regulates mitochondrial homeostasis [22,23]. Using bioinformatics analysis, we identified ten amino acids that are similar between Pink1 and Opa1, GLQRMVLVDL (Figure 1A). This short domain is highly conserved among multiple species (Figure 1B,C), and only conservative amino acid substitutions were observed in Opa1 (L to I). In addition, this sequence is unique to Opa1 (not present in other proteins, data not shown), and it is exposed in Pink1.

### 2.2. Identify Pink1 Probe Interaction Sites Using Cross-Linking and Mass Spectrometry

Pink1 is a mitochondrial serine/threonine kinase encoded by the *PINK1* gene, whose kinase domain is localized in the OMM. This kinase domain is accessible from the cytoplasm and is activated in cells upon mitochondrial membrane depolarization, but the mechanism of activation remains unknown. Pink1 is composed of a mitochondrial targeting sequence located at the N-terminal, a transmembrane helix, an N-terminal regulatory domain, a serine/threonine kinase domain that is constituted by the N-lobe and C-lobe, and the C-terminal portion that has been associated with the regulation of its kinase activity (Figure 2).

Molecular docking techniques are ideal for gaining insight into the structure of binding sites and the affinities of various ligands towards the target protein. We applied peptide-protein docking studies to predict the site of interaction of the peptide with the human Pink1 (hPink1) protein. First, we used the Chou & Fasman Secondary Structure Prediction Server (CFSSP, http://www.biogem.org/tool/chou-fasman/, accessed on 1 April 2022) [25] to perform a secondary structure prediction of the cargo peptide. The structural prediction indicated that this region consists mainly of a helical secondary structure. Next, the Pymol building tool was used to prepare the peptide that was then docked with the structure of the hPink1 protein that was predicted by AlphaFold [26] (AlphaFold predicted model: Q9BXM7) using the PatchDock online docking server (https://bioinfo3d.cs.tau.ac.il/PatchDock/php.php, 1 April 2022). The server used a geometry-based molecular docking algorithm as a scoring function [27,28]. We selected the top peptide conformation scores produced by PatchDock. The docking analysis results were analyzed, and 3D graphics were generated using PyMOL (Appendix A). The docking study showed that the peptide binds within a binding pocket formed by helices of the C-lobe and the CTD together with random coils of the N-lobe.

To further investigate the docking result and to validate the peptide-protein interaction sites, we applied crosslinking coupled with mass spectrometry (CLMS). Here, two water-soluble cross-linkers were used for the CLMS: 4-(4,6-dimethoxy-1,3,5-triazin-2-yl)-4-methyl-morpholinium chloride (DMTMM), and bis(sulfosuccinimidyl)suberate (BS3). DMTMM is a hetero-bifunctional amine-to-carboxylic acid crosslinker, and BS3 is a homo-bifunctional amine-to-amine crosslinker, which reacts primarily with the ε-amino group of lysine residues in proteins.

Based on the identified peptide, we developed the Pink1 molecular probe for cross-linking studies. A probe with an extra lysine residue located adjacent to the bioactive cargo was developed to better locate and identify the interactions between the protein and the bioactive cargo (CVP-198, Appendix A). Considering the length of the crosslinkers (15 Å for DMTMM and 11 Å for BS3), the side chain of the target residues (e.g., lysine, aspartic acid, and glutamic acid), and backbone dynamics, we assumed that these residues within a Cα-Cα distance of up to 35 Å would be preferentially crosslinked [29]. The additional lysine adjacent to the bioactive cargo domain of the peptide was identified to be particularly reactive. As this residue is localized near the cargo of the peptide, this is a good access point to a distinct role of the bioactive part of the peptide.

We identified eight sequences of peptide pairs reporting a cross-link between the protein Pink1 sequence and the peptide CVP-198 using a BS3 cross-linker. The lysine that was introduced adjacent to the peptide cargo was the reactive lysine in all of the cases at the peptide sequence, while seven different lysines were identified on Pink1 proteins, residues 137, 186, 266, 319, 496, 547, and 555. Using DMTMM cross-linker, two peptide pairs reporting a cross-link between the protein and the peptide were identified. The lysine that was introduced adjacent to the peptide cargo with glutamic acid (residue 155) of Pink1, as well as aspartic acid of the cargo peptide (residue 9), with lysine (residue 137) of Pink1. Of the nine intermolecular cross-linked peptide-protein pairs identified using the CLMS approach, 89% (8 out of 9) are consistent with the docking results analysis. Lysine (residue 496) is not compatible with the model structure of Pink1. However, all the other identified peptide-protein interaction crosslinks fit well with the model structure of Pink1. The combination of the crosslinking results and bioinformatics docking analysis demonstrates a high degree of complementarity between these two orthogonal approaches (Figure 3A,B).

### 2.3. Develop Pink1 Fluorescent Molecular Probe

Based on the identified amino acid sequence, we also developed a Pink1 fluorescent molecular probe. Cell entry of the probe was facilitated by the conjugation of cell penetrating peptides (CPP), short peptides that are able to translocate across the cell membrane and further assist the intracellular delivery of bioactive cargos [30,31]. Next, a cysteine residue was added to the peptide sequence to introduce fluorescent labeling. The cysteine residue allows site-specific conjugation due to the specific reaction of the cysteine thiol (–SH) with widely available maleimide-conjugated dyes. Finally, a Boron-dipyrromethene (BDP) dye with Excitation/Emission (Ex/Em) 628/642 was used. BDP dyes have a relatively long fluorescence lifetime, high quantum yields, and they have been used for many applications, including the direct observation of dynamic organelle fusion processes, in vivo site-specific imaging, and colocalization analysis [32].

We developed CVP-206, which is composed of the bioactive peptide (aka cargo, GLQRMVLVDL), CPP (TAT carrier), a short spacer for enhancing the peptide activity and flexibility, and a cysteine residue for conjugation to a fluorophore (BDP) (Appendix A). To confirm the interaction between Pink1 and the peptide, we monitored their colocalization by confocal microscopy in the H9c2 rat cardiomyoblast cell line, which was treated with hydrogen peroxide (H_2_O_2_) that induced cellular apoptosis, reactive oxygen species (ROS) production, mitochondrial structure disruption, and activation of key signaling proteins in the mitochondrial apoptotic pathway [33]. MitoTracker was used to confirm the localization of Pink1 to the mitochondrial membrane (Appendix A). H_2_O_2_ treatment enhanced the localization of Pink1 at the OMM and the colocalization of Pink1 with the peptide, CVP-206, whereas this enhancement was significantly reduced in control cells (Figure 4).

### 2.4. Design and Synthesis of Backbone Cyclic Mimetic Library

Linear peptides demonstrate several drawbacks, such as high conformational flexibility (which may result in low receptor selectivity), poor stability (due to rapid digestion by proteolytic enzymes), and limited permeability. Cyclic peptides can overcome linear peptide limitations, demonstrating better biological activity and selectivity. Yet, the biologically active conformation of a peptide is selected and/or induced when interacting with its protein partner, as demonstrated by the “Induced fit” model [34], a development of the “Lock and Key” model [35]. Cyclic peptides that are conformationally constrained may result in limited bioactivity, as their induced fit upon binding to a target is limited. Consequently, to identify a bioactive cyclic peptide(s) with desirable biological activity, a library of cyclic peptides should be screened [36].

We used backbone cyclization that enables the development of cyclic peptides without utilizing the residues that are part of the natural linear peptide, which are typically critical for binding and biological function, keeping the natural amino acid functional groups to support biological activity [37,38,39]. The design of the backbone cyclic peptide library was based on the identified bioactive linear peptide, GLQRMVLVDL. Additional lysine residue was introduced to the peptide to perform an N-backbone to end cyclization without using any of the original bioactive peptide pharmacophores. The position of the cyclization in the peptide chain was determined to include only the cargo, as once the peptide reaches the intracellular target, the CPP has finished its purpose and may be degraded for cargo release. To increase the number of possible conformations that can potentially support peptide activity, a library of backbone cyclic peptides that contain several variable methylene linker chains has been developed.

Peptide analogous, including precyclic and backbone cyclic peptides, were developed using Fmoc solid phase peptide synthesis (SPPS) standard protocols based on the identified bioactive sequence. The cyclization was performed by connecting a dicarboxylic linker to the amino terminus of the peptide by the addition of a lysine side chain that was protected with N-methyltrityl (Mtt) [40], a protecting group that can be deprotected selectively under acid labile conditions [41,42], which is appropriate for Fmoc SPPS protocols. All backbone peptides analogous to the linear peptide have the same primary sequence but differ in the bridge size. It was demonstrated that the amino and the carboxy terminus have an important role in peptide stability and bioactivity, thus the amino terminus was protected throughout the cyclization step, and the carboxy terminus by C-terminal amidation that reduced the overall charge of a peptide. In addition, it increased the stability of the peptide as it generates a closer mimic of the native protein. Coupling was done using the DIC (carbodiimide)/Oxyma (ethyl 2-cyano-2-(hydroxyimino)acetate) approach, as it is a base-free condition known to reduce epimerization [43]. Additionally, DIC is stable at high temperature (90 °C) [44]. The cyclization step has been done via an amide bond to a dicarboxylic acid spacer attached to the N-terminus of the peptide by SPPS methodology, by means of “on-resin cyclization”. Final acidic cleavage followed by high-performance liquid chromatography (HPLC) purification produced the desired peptides. The final library contained five peptides: linear, precyclic, and backbone cyclic peptides, all of them derived from the identified linear sequence to mimic its bioactivity effect (Figure 5 and Table 1; for full characterization see Appendix A).

### 2.5. Binding of Linear and Backbone Cyclic Peptides

We determined the peptides bound to *Tribolium castaneum* (Tc) Pink1, the most active orthologue of Pink1 [45], using field-effect biosensing (FEB) technology, a label-free biophysical detection method in which the results are monitored in real-time, as we have done previously [14,17,46]. While, in general, cyclization of a linear peptide can lead to increased affinity for the target protein, it was demonstrated that constrained ligands can possess either enhanced, reduced, or unaltered binding over their linear counterparts [47].

CVP-198 is the linear peptide bound to TcPink1 protein in vitro with K_D_ = 130 µM; while the precyclic peptides, CVP-201 and CVP-202, which imposed some conformational constraints, demonstrated similar binding (K_D_ = 126 µM and K_D_ = 143 µM, respectively). Finally, the cyclic peptide CVP-199 demonstrated the highest binding affinity (K_D_ = 58 µM), under the same experimental conditions (Figure 6 and Table 2). Interestingly, the second cyclic peptide, CVP-200, showed low binding to the target (K_D_ = 139 µM), demonstrating the importance of developing a library with several cyclic peptides to screen their conformational space.

### 2.6. Stability of the Biomolecular Probes

Enzymatic degradation is known to affect the half-lives of peptides. As peptides are being evaluated for their use in basic research and therapeutic applications, it is essential to validate the stability of the designed sequence in the given peptide. To assess the impact of peptide cyclization on pharmacokinetic parameters, the metabolic stability of the linear peptide as well as the more active backbone cyclic peptide were measured.

We tested the stability of the peptides using trypsin, an endopeptidase with specificity for cleaving the amide bond between a cationic amino acid and the next amino acid in the C-terminal direction, which is found in the digestive systems of many vertebrates, and it is commonly used as an important enzymatic reagent in biochemistry and biology.

In general, peptides are protease-susceptible, and cyclization is a well-established technique for stabilizing peptides and improving their pharmacokinetic profiles. One of the benefits of the cyclic structure is the resistance to hydrolysis by exopeptidases due to the lack of amino and/or carboxyl termini. Cyclic peptides can even be resistant to endopeptidases as their structure is less flexible than linear peptides. As expected, the linear peptide, CVP-198, was rapidly degraded by trypsin, which showed high degradation of the peptide (~70%) after 30 min of incubation at 37 °C with trypsin, while the backbone cyclic peptide, CVP-199, had prolonged intact peptide presence (only 45% degradation after 30 min), suggesting that the cyclization protects the peptide from degradation. After approximately 120 min, about 95% degradation of the linear peptide occurred after incubation at 37 °C with trypsin. Proteolytic cleavage of the cyclic peptide was much slower, with less than 75% of the cyclic peptide cleaved in 120 min. The cyclic peptide was found to be significantly more stable than the linear peptide (Figure 7).

## 3. Discussion

Pink1 is a mitochondrially targeted serine/threonine kinase which controls the specific elimination of dysfunctional mitochondria. It accumulates on defective mitochondria, eliciting the translocation of Parkin from the cytosol to mediate the clearance of damaged mitochondria via mitophagy. Pink1 is involved in many biological metabolic processes, and there is much evidence that altered Pink1/Parkin related mitophagy may be involved in the pathogenesis of various human diseases. In addition, Pink1 is rapidly turned over under basal conditions, and its expression level is very low. While Pink1 has a critical role in physiological and pathological conditions, there are very limited tools to study its role in vitro and in vivo.

We developed a set of molecular probes to further study the role of Pink1 in health and disease (Figure 8). Initially, we rationally designed a peptide that targets Pink1 protein as a tool to explore Pink1 functions, and we conjugated CPP to the bioactive peptide to deliver it inside the intact cells. Next, using cross linker fusion, we mapped the three-dimensional (3D) structure of the interaction site of the target protein with the peptide, which aligned with the docking prediction. In addition, using straightforward chemistry, we conjugated fluorescent dye to the bioactive peptide and demonstrated that it is colocalized with the target protein in cells. Finally, we used the backbone cyclization approach to overcome linear peptide limitation, mainly rapid proteolysis and inadequate membrane permeability, and developed peptidomimetics with increased stability and improved binding efficacy compared to their linear counterparts.

In summary, we present a step-by-step guideline to transform bioactive peptides into molecular probes, generating unique research tools for basic research and drug discovery. We demonstrate, using straightforward chemical synthesis approaches, how to optimize a bioactive linear peptide into designated probes. Modifications, such as attachment to cell-penetrating peptide, dye conjugation, cross-linker fusion, and cyclization, all result in a promising class of tools that can advance the revealing of basic research mechanisms and unknown signaling pathways, as well as for drug discovery in which peptides demonstrate extremely tight binding to their targets, high specificity, and low toxicity. The methods presented herein are general and common and can be used by many laboratories to custom identify bioactive peptides to be more specific and effective bioactive tools to advance basic research as well as therapeutic leads.

## 4. Materials and Methods

### 4.1. Sequence Alignments

Sequences from different species were aligned using the LALIGN server (accessed on 1 April 2022), using Pink1 proteins (Homo sapiens (Q9BXM7), Mus musculus (Q99MQ3), Rattus norvegicus (B5DFG1)), Opa1 proteins (Homo sapiens (O60313), Mus musculus (P58281), Rattus norvegicus (Q2TA68), Gallus gallus (Q5F499), Danio rerio (Q5U3A7), and Drosophila melanogaster (A1Z9N0)).

### 4.2. Peptide Synthesis

In brief: Peptides were chemically synthesized using a fully automated peptide synthesizer (Syro I, Biotage, Uppsala, Sweden) on solid support by following the solid-phase peptide synthesis (SPPS) methodology [48] with a fluorenylmethoxycarbonyl (Fmoc)/tert-Butyl (tBu) protocol. The cyclic peptides were synthesized with a modified lysine whose side chain was protected with N-methyltrityl (Mtt), a protection group that can be deprotected selectively using acid labile conditions [40]. After completion of the synthesis of the linear peptide, an anhydride spacer was coupled to the N-terminal amino group and cyclization was performed using amide bonds between the moiety linker at the backbone N-terminus and an epsilon amino on the side chain of a lysine residue [37]. The final cleavage and side chain deprotection were done manually. Peptides were analyzed by analytical reverse-phase high-pressure liquid chromatography (RP-HPLC) (1260 Infinity II LC System, Agilent, Santa Clara, CA, USA) and matrix assisted laser desorption/ionization mass spectrometry (MALDI-MS) (autoflex^®^ maX, Bruker, Billerica, MA, USA) and purified by preparative RP-HPLC (1260 Infinity II LC System, Agilent, Santa Clara, CA, USA).

Further details: All commercially available solvents and reagents were used without further purification. Dichloromethane (DCM), Piperidine, Diethyl ether, N,N Diisopropylethylamine (DIEA), Trifluoroacetic acid (TFA), and Water (HPLC grade) were purchased from Bio-Lab (Jerusalem, Israel); Acetonitrile (ACN) (HPLC grade) was purchased from J.T. Baker (Poland); Acetic anhydride was purchased from Daejung (Gimhae-si, Korea); Triisopropylsilane (TIS), and Succinic anhydride were purchased from Acros organics (Branchburg, NJ, USA); Dithiothreitol (DTT) was purchased from Fisher Bioreagents (Ottawa, ON, Canada); Dimethylformamide (DMF) was purchased from Carlo Erba (Val De Reuil, France); Oxyma Pure was contributed by Luxembourg Bio Technologies Ltd. (Ness Ziona, Israel); N,N-Diisopropylcarbodiimide (DIC) was purchased from Angene International Limited (Nanjing, China); Glutaric anhydride was purchased from Alfa Aesar (Chaoyang, China); Benzotriazole-1-yl-oxy-tris-pyrrolidinophosphonium hexafluorophosphate (PyBOP) was purchased from GL Biochem Ltd. (Shanghai, China); Fmoc Rink amide MBHA resin was purchased from AnaSpec (substitution 0.67 mmol/g, Fremont, CA, USA); Fmoc-protected amino acids were purchased from Ontores Biotechnologies (Hangzhou, China). Side chains of the amino acids used in the synthesis were protected as follows: tert-Butyloxycarbonyl (BOC) (Lys/Trp), tert-butyl (tBu) (Ser/Thr/Tyr/Glu), t-butyl ester (OtBu) (Asp), 2,2,4,6,7-Pentamethyldihydrobenzofuran-5-sulfonyl (Pbf) (Arg), 4-methyltrityl (Mtt) (Lys), and triphenylmethy (Trt) (Asn/Cys/Gln/His).

Peptides were chemically synthesized using a fully automated parallel peptide synthesizer, Syro I (Biotage, Uppsala, Sweden), on solid support following the fluorenylmethoxycarbonyl (Fmoc)/tert-Butyl (tBu) method. Fmoc deprotection was performed in two steps: 3 min and 12 min, both at 75 °C using piperidine (40%) in DMF solution. Coupling reactions were performed by repetition of the following cycle conditions: 45 min, 75 °C, with DIC (0.2 M) in DMF, Oxyma Pure (0.2 M) in DMF, and amino acid (0.2 M) in DMF. Whenever necessary, coupling and Fmoc deprotection steps were monitored using small cleavage. Anhydride coupling was carried out by the repetition of the following cycle conditions: 30 min at room temperature, using anhydride (10 eq)/ DIEA (10 eq)/ peptide (1 eq) in DMF. Mtt deprotection was carried out using TFA/TIS/DCM (1:5:94) and the reaction was done for 5 min, three times, at room temperature. Cyclization was performed manually in a chemical hood using DCM and PyBOP/DIEA (5:10) for 60 min at room temperature.

Peptide cleavage from the resin and deprotection of the amino acid side chains were carried out with a pre-cooled mixture of TFA/TIS/H_2_O solution (90:2.5:2.5 *v*/*v*/*v*) for three hours at room temperature. For peptides that have Cys in the sequence, the cleavage cocktail was TFA/TIS/H_2_O/DTT solution (94:1:5:50 *v*/*v*/*v*/(mg/mL)) and the reaction was done for three hours at room temperature. The resin was removed by filtration, and the solvents were evaporated by a stream of compressed air. The crude products were precipitated with diethyl ether, collected by centrifugation, dissolved in CH_3_CN/H_2_O (30:70) and lyophilized.

Small Cleavage. A small amount of resin was treated with a pre-cooled mixture of TFA/H_2_O/TIS (95:2.5:2.5) and the reaction was carried out for 60 min at room temperature. The resin was removed by filtration, and the solvents were evaporated by a stream of compressed air. The residue was dissolved in CH_3_CN/H_2_O (30:70). The filtrated solution was analyzed by HPLC and/or MS.

Products were analyzed by analytical reverse-phase high-pressure liquid chromatography (RP-HPLC) 1260 Infinity II LC System equipped with: G7129A 1260 vialsampler, G7111B 1260 quaternary pump, G7115A 1260 DAD (Diode Array Detector) WR, G1364C 1260 FC-AS, G1330B 1290 thermostat, from Agilent (Santa Clara, CA, USA) using a Luna 5 µm C18(2) 100 Å (250 × 4.6 mm) column (Phenomenex, Torrance, CA, USA) at 1 mL/min. The solvent systems used were A (H_2_O with 0.1% TFA) and B (CH_3_CN with 0.1% TFA). A linear gradient of 5–95% B in 45 min was applied and the detection was at 214 nm and 254 nm. The synthesis products were purified using a Luna 5 µm C18(2) 100 Å (250 × 10 mm) column (Phenomenex, Torrance, CA, USA) at 4.7 mL/min. The solvent systems used were A (H_2_O with 0.1% TFA) and B (CH_3_CN with 0.1% TFA). For separation, a linear gradient of 5–95% B in 45 min was applied and the detection was at 214 nm and 254 nm.

### 4.3. Conjugating Dye to the Peptide

In brief: Dye was covalently attached by the Sulphur of the Cys in the peptide with Tris buffer solution at room temperature.

BDP 630/650 maleimide was purchased from Lumiprobe (Hunt Valley, MD, USA); Tris HCl was purchased from Fisher Bioreagents (Shanghai, China); Tris base was purchased from Fisher Bioreagents (Pittsburgh, PA, USA). The peptide CVD-205 (CH_3_-C(O)-CGLQRMVLVDLKGGYGRKKRRQRRR-NH_2_) (Appendix A) was dissolved in Tris buffer (100 mM (pH 7.4)) to 4 mM. The dye was dissolved in ACN to 3 mM. The dye solution was added to the peptide solution in equal volumes with proper shaking at 700 rounds per minute (rpm) overnight at room temperature.

### 4.4. Molecular Docking

The protein structure was input as the receptor molecule, and the peptides were used as the ligands in the PatchDock webserver [28]. The final clustering was selected on the basis of the root-mean-square deviation (RMSD) value. The best protein–peptide complex was selected from among the top ten conformers based on energy scoring. This web tool was employed to obtain rigid protein–peptide docking.

### 4.5. Stability Test of Linear and Cyclic Peptide

In brief: The stability studies of the peptides were performed by dissolving the peptides in Tris buffer and incubating them with trypsin solution at 37 °C using HPLC.

Further details: Trypsin from porcine pancreas was purchased from Sigma-Aldrich (Saint Louis, MO, USA). Peptide (2 mg) was dissolved in Tris buffer (800 µL, 50 mM (pH 8.0)). The peptide solution was mixed with 1 μL of trypsin solution (1 mg/mL in 50 mM Tris buffer (pH 8.0)). The peptide was incubated at 37 °C and samples (90 μL) were taken at 0, 5, 30, 60, and 120 min. The samples were mixed with TFA/CH_3_CN/H_2_O (2%/5%/93%, 90 µL). The peptides were analyzed by analytical reverse-phase high-pressure liquid chromatography (RP-HPLC) 1260 Infinity II LC System from Agilent, (Santa Clara, CA, USA) using a Luna 5 μm C18(2) 100 Å (250 × 4.6 mm) column (Phenomenex, Torrance, CA, USA) at 1 mL/min. The solvent systems used were A (H_2_O with 0.1% TFA) and B (CH_3_CN with 0.1% TFA). A linear gradient of 5–50% B in 15 min was applied, and the detection was at 214 nm and 254 nm.

### 4.6. Protein Expression and Purification

#### 4.6.1. Bacterial Protein Expression

The two recombinant proteins, human Pink1 (hPink1) and *Tribolium castaneum* Pink1 (TcPink1), were expressed using the *Escherichia coli (E. coli)* Rosetta bacterial expression system. The hPink1 plasmid construct, pET-21a(+)/Maltose binding protein (MBP)-His-Pink1, consists of the partial AA (125–581) sequence of hPink1, which was purchased from Addgene (Watertown, MA, USA). The TcPink1 plasmid construct, pMal4c Pink1 (*Tribolium castaneum*) DU34701 consists of full-length AA sequence (1–570) was purchased from the MRC Protein Phosphorylation and Ubiquitylation Unit, Dundee University (Dundee, UK).

All constructs were transformed into *E. coli* XL1-Gold strain Stratagene (San Francisco, CA, USA) by using the conventional heat shock method, followed by cultivation and plasmid DNA purification using the QIAprep minispin kit (QIAGEN, Hilden, Germany). All plasmids were further sequenced and validated by Sanger sequencing (Macrogen via IDT). The plasmids were stored at −20 °C until use for protein expression.

For protein expression, the validated plasmid constructs of hPink1 and TcPink1 were transformed into *E. coli* Rosetta (DE3) strain cells (Novagen, Madison, WI, USA) by the heat shock method. The cells were plated on Lysogeny broth (LB)-agar supplemented with Ampicillin + Chloramphenicol and allowed to incubate overnight at 37 °C. A single colony was inoculated into 2 mL LB media supplemented with the appropriate antibiotics and glucose (0.2%) and incubated overnight at 37 °C with shaking at 200 rpm. Upon reaching OD_600_ of 0.5–0.8, the expressions of hPink1 and TcPink1 were induced in bacterial cells with isopropyl β-D-1-thiogalactopyranoside (IPTG) (0.25 mM), and the culture was incubated at 16 °C for 14–16 h with proper shaking at 200 rpm. Cells were then harvested by centrifugation (SL 16R, Thermo Fisher, Waltham, MA, USA) (20 min, 4000 rpm, 4 °C). The pellet was re-suspended in ice-cold lysis (L) buffer (L-MBP buffer: 50 mM Tris-HCl (pH 7.5), 150 mM NaCl, 1 mM Ethylenediaminetetraacetic acid (EDTA), 1 mM ethylene glycol-bis(β-aminoethyl ether)-N,N,N′,N′-tetraacetic acid (EGTA), 5% (*v*/*v*) glycerol, 1% (*v*/*v*) Triton X-100, 0.1% (*v*/*v*) 2-mercaptoethanol, 1 mM benzamidine, 0.1 mM phenylmethylsulfonyl fluoride (PMSF), and complete TM EDTA-free protease inhibitors (Roche, Basel, Switzerland)).

#### 4.6.2. Protein Purification

After re-suspension in lysis buffer, cells were further lysed via EmulsiFlex^®^-C3, high pressure homogenizer (AVESTIN, Ottawa, ON, Canada) (15,000 psi, 10 min, 4 °C). To separate the soluble protein from the cell debris, the cell lysate was subjected to centrifugation in the Avanti J-E centrifuge system (Beckman Coulter, Brea, CA, USA) (30,000× *g*, 45 min, 4 °C). The soluble supernatant of the lysate was filtered and loaded onto an immobilized-amylose affinity chromatography column (Amylose resin, New England Bio Labs, Ipswich, MA, USA) using an automated fast liquid chromatography (FPLC) AKTA system (GE Healthcare, Chicago, IL, USA). The affinity column bound hPink1 or TcPink1 proteins were eluted from the column using the elution buffer containing maltose (12 mM).

The OD of eluted fractions were monitored in a nano-spectrophotometer at 280 nm and fractions containing hPink1 were pooled together and concentrated to 1.3 mg/mL in 3 mL using a spin concentrator (vivaspin 500, Sartorius, Goettingen, Germany), with the appropriate cut off (100 kDa). The sample was loaded onto size-exclusion chromatography (SEC) with a Superdex 200 prep grade (16/700) GL column (GE Healthcare, Chicago, IL, USA) using the FPLC AKTA system, preequilibrated with Tris-HCl buffer (50 mM Tris-HCl (pH 7.5), 150 mM NaCl, 0.1 mM EGTA, 5% (*v*/*v*) glycerol, 0.03% (*v*/*v*) Brij-35, and 0.1% (*v*/*v*) 2-mercaptoethanol).

Peak fractions were analyzed in sodium dodecyl sulfate-polyacrylamide gel electrophoresis (SDS-PAGE) gels (10%) and stained with Coomassie blue. The Western blot analysis for TcPink1 was done with a MBP mouse monoclonal antibody, sc-13564 (Santa Cruz Biotechnology, Dallas, TX, USA). To identify the recombinant hPink1, Western blot analysis was done with Pink1 rabbit monoclonal antibody, 6946S (Cell Signaling Technology, Beverly, MA, USA). Secondary antibodies, Anti-mouse DyLight™ 680, 5470S (Cell Signaling Technology, Beverly, MA, USA) and Anti-rabbit DyLight™ 800, 5151S (Cell Signaling Technology, Beverly, MA, USA), were used in the analysis that was done using Odyssey^®^ CLx (LI COR Biosciences, Lincoln, NE, USA). Relevant fractions containing Pink1 were flash frozen in liquid nitrogen. Purified Pink1 proteins were stored at −80 °C in affinity chromatography elution buffer or in SEC elution buffer. Next, the proteins were dialyzed into the proper buffer as needed in each protocol.

### 4.7. Peptide Binding to Protein, In Vitro

In brief: Protein was immobilized/cross-linked into the carboxyl group present on the activated graphene biosensor chip. The analyte is applied in solution to the chip and when an interaction occurs, an alteration in the current (I) is measured and recorded in real-time throughout the experiment. The entire cycle of the experiment starts with calibration using PBS (pH 7.4) to record the baseline equilibration response. Followed by the association step by adding analyte (50 µL) at the desired concentrations in PBS (pH 7.4). After the experiment, data at the concentration points of the analytes were exported from 3 transistors, averaged, and any background drift recorded in PBS was subtracted.

Further details: N-hydroxysulfosuccinimide sadium salt (Sulfo-NHS) was purchased from Biosynth Carbosynth (Compton, UK); 1-(3-Dimethylaminopropyl)-3-ethylcarbodiimide hydrochloride (EDC-HCl) was purchased from Alfa Aesar (Kandel, Germany); Quench 1 (3.9 mM amino-PEG5-alcohol in PBS (pH 7.4)) and Quench 2 (1 M ethanolamine (pH 8.5)) were purchased from Cardea (San Diego, CA, USA); 2-(N-morpholino)-ethanesulfonic acid (MES) was purchased from Sigma-Aldrich (Saint Louis, MO, USA); PBS X 10 (pH 7.0) was purchased from Hylabs (Rehovot, Israel).

Binding data of the peptides to immobilized protein in vitro was gathered using an AGILE Dev Kit label-free binding assay (Cardea, San Diego, CA, USA). Following the standard protocol from the manufacturer. The capture molecules were bound to the chip with a zero length linker (EDC/sulfo-NHS). EDC (2 mg) and sulfo-NHS (6 mg) were used in MES buffer (1 M (pH 6.0)) for 15 min to covalently attach the amine of the protein to the carboxyl on the chip. The protein solution (500 nM) was incubated with the chip for 15 min. Next, Quench 1 followed by Quench 2 were applied serially for 15 min each to quench remaining unoccupied binding sites on the chip. After a rinse in PBS, base-line current levels for the chip were recorded for at least 5 min. Next, the PBS was aspirated, and a droplet of the tested analyte (50 μL) was applied to the chip and the change in the sensor chip readout was recorded, after that the analyte was aspirated and the chip was rinsed with PBS. Additional measurements were performed using varying analyte concentrations. After data were gathered, the responses of the sensors on a single assay chip were averaged, and any background drift recorded in PBS was subtracted. A Hill fit plot was used to determine a K_D_. The K_D_ values were calculated by GraghPad Prism 9, statistical analysis software. Data presented as mean ± SEM of all measurements. All samples were identical prior to allocation of treatments.

### 4.8. Crosslinking Study

#### 4.8.1. Protein Cross-Linking

In brief: The recombinant protein and the peptide were incubated together with a crosslinker. The reaction was stopped with a quencher. The proteins were precipitated collected and trypsinased, the outcome was analysed by liquid chromatography mass spectrometry (LCMS).

Further details: Suberic acid bis(3-sulfo-N-hydroxysuccinimide ester) sodium salt (BS3), 4-(4,6-Dimethoxy-1,3,5-triazin-2-yl)-4-methylmorpholinium chloride (DMTMM), and iodoacetamide were purchased from Sigma-Aldrich (Saint Louis, MO, USA); Sequencing Grade Modified Trypsin (lyophilized) was purchased from Promega (Fitchburg, WI, USA); Empore™ SPE Disks matrix active group C18 was purchased from Merck (Darmstadt, Germany); 4-(2-hydroxyethyl)-1-piperazineethanesulfonic acid (HEPES) was purchased from Fisher Bioreagents (Pittsburgh, PA, USA); Dimethyl Sulfoxide (DMSO) was purchased from Bio-Lab (Jerusalem, Israel).

A mixture solution of purified protein and peptide was prepared in HEPES buffer (50 mM HEPES, 150 mM NaCl (pH 8.0)). Protein TcPink1’s final concentration was 10 μM and hPink1’s final concentration was 2 μM. The peptide was added at a 1:10 or 1:100 protein-to-peptide ratio, and the mixture incubated with agitation for 20 min at room temperature. For BS3 cross-linking, a solution of BS3 in DMSO (150 mM) was used, which was diluted with the proteins to a final BS3 concentration of 1 mM. The cross-linking reaction was incubated at room temperature for 30 min with agitation. The reaction was quenched by adding ammonium bicarbonate to a final concentration of 30 mM. For DMTMM cross-linking, a solution of DMTMM in HEPES buffer (100 mM) was used, which was diluted with the proteins to a final DMTMM concentration of 7 mM. The cross-linking reaction was incubated at room temperature for 30 min with agitation at 600 rpm. The reaction was quenched by adding ammonium bicarbonate (100 mM).

#### 4.8.2. Mass Spectrometry

The proteins were precipitated in acetone at −80 °C for 1 h, followed by centrifugation at 13,000× *g*. The pellet was resuspended in urea (20 μL, 8 M) with DTT (10 mM). After 30 min, iodoacetamide was added to a final concentration of 25 mM and the alkylation reaction proceeded for 30 min in the dark. The urea was diluted by adding 250 μL digestion buffer (25 mM TRIS (pH 8.0); 10% acetonitrile), trypsin (Promega, Fitchburg, WI, USA) was added at a 1:100 protease-to-protein ratio, and the protein was digested overnight at 37 °C with agitation. Following digestion, the peptides were desalted on C18 stage-tips and eluted by 55% acetonitrile. The eluted peptides were dried in a SpeedVac, reconstituted in formic acid (0.1%), and measured in a mass spectrometer. The samples were analyzed by a 60 min 0–40% acetonitrile gradient on a liquid chromatography system (Acquity M UPLC, Waters, Milford, MA, USA) coupled to a Q-Exactive Plus mass spectrometer (Thermo Fisher, Waltham, MA, USA). The RAW data files from the mass spectrometer were converted into MGF format in a Proteome Discoverer (Thermo Fisher, Waltham, MA, USA), which was the input format for the analysis pipeline. The method parameters of the runs were as follows: data-dependent acquisition; full MS resolution, 70,000; MS1 AGC target, 1e6; MS1 maximum IT, 200 ms; scan range, 450–1800; dd-MS/MS resolution, 35,000; MS/MS AGC target, 2e5; MS2 maximum IT, 300 ms; loop count, top 12; isolation window, 1.1; fixed first mass, 130; HCD energy (NCE), 26; MS2 minimum AGC target, 800; charge exclusion: unassigned, 1, 2, 3, 8, >8; peptide match, off; exclude isotope, on; and dynamic exclusion, 45 s.

#### 4.8.3. Identification of Cross-Links

We used an established search application [49]. The sequence database included TcPink1, hPink1 and the peptide, CVP-198. The false discovery rate (FDR) was estimated from decoy-based analysis, which repeated the identification analysis 20 times with an erroneous cross-linker mass. For example, in the case of BS3 cross-linking, which has a true cross-linker mass of 138.0681 Da, erroneous cross-linker masses of 138.0681 * N/138 Da were used, where N = 160, 161, 162, …, 179. This led to bogus identifications with fragmentation scores that were generally much lower than the scores obtained with the correct cross-linker mass (see histograms in Appendix A). For the identification of true cross-links, we set the threshold on the fragmentation score according to the desired FDR value. For example, a threshold of 0.65 on the fragmentation score of the BS3 cross-linked CVP-198 and hPink1 datasets, gave 75 cross-links above the threshold in the true analysis and a median of 1 cross-link in a typical decoy run (Appendix A). We therefore estimate the corresponding FDR to be about 1 in 75, or ~2%. The final thresholds used: BS3 cross-linked mixtures, 0.65, and DMTMM cross-linked mixtures, 0.7.

### 4.9. Cells Study

In brief: H9c2 cells were treated by hydrogen peroxide (H_2_O_2_) followed by the addition of dye conjugated peptide, and the cells were fixed and permeabilized. Immunocytochemistry was done with anti-Pink1 and a secondary antibody. The control experiment was done without the H_2_O_2_ stress. The Pink1 location on the mitochondria was checked by adding MitoTracker to the cells, which were fixed and permeabilized. The cells in all conditions were analyzed by a confocal microscope.

Further details: H9c2 cell line derived from embryonic rat heart tissue (CRL-1446) was purchased from the American Type Culture Collection (ATCC, Gaithersburg, MD, USA); Dulbecco’s modified Eagle’s high glucose medium (DMEM), fetal bovine serum (FBS), sodium pyruvate, penicillin–streptomycin (Pen–Strep) (10×) and combined antibiotic solutions were purchased from Biological Industries (Beit-Haemek, Israel); Sodium hydrogen carbonate was purchased from Daejung (Siheung-Si, Korea); Glass bottom dish 35 mm with 10 mm bottom well #1.5 glass were purchased from Cellvis (Mountain View, CA, USA); Hydrogen peroxide (30%) was purchased from Fisher Bioreagents (Pittsburgh, PA, USA); Formaldehyde solution phosphate buffered (4%) was purchased from Fisher Chemical (Loughborough, UK); MitoTracker Red CM-H2Xros was purchased from Thermo Fisher Scientific (Waltham, MA, USA).

H9c2 cells were grown in standard DMEM containing FBS (10%), penicillin-streptomycin (100 U/mL–100 ug/mL), sodium pyruvate (1%), and sodium hydrogen carbonate (0.15%). Cells were plated in glass bottom 35 mm plates at a density of 10,000 cells per well and incubated overnight in a cell culture incubator supplied with 5% CO_2_ at 37 °C. A stock solution of the colored peptide CVP-206 (1 mM) was prepared in DMSO. Cells undergoing oxidative stress were incubated in serum-free DMEM media with H_2_O_2_ (1.6 mM) and incubated for 0.5 h in a cell culture incubator supplied with 5% CO_2_ at 37 °C. After washing with serum-free media, a peptide with dye, CVP-206 (1 µM) or MitoTracker Red CM-H2Xros (300 nM) in serum-free media, was applied to the cells and incubated for an additional 0.5 h in a cell culture incubator supplied with 5% CO_2_ at 37 °C. The cells were washed with PBS and fixed with paraformaldehyde (4%) in PBS (pH 7.3) for 15 min. Cells washed with PBS 0.2% Tween 20 and undergo permeabilization with PBS 0.3% Triton X-100 for 15 min. Next, cells were washed with PBS 0.2% Tween 20 and blocked with horse serum (1%) in PBS 0.2% Tween 20 for 1 h at room temperature. Anti-Pink1 antibody (AB-ab186303, Abcam, Cambridge, UK) 1:100 in horse serum (1%) in PBS 0.2% Tween 20 was applied to the cells for 1 h at room temperature. Next, Goat anti-Mouse IgG H&L (Alexa Fluor 488) preadsorbed (ab150117, Abcam, Cambridge, UK) 1:1000 in horse serum (1%) in PBS 0.2% Tween 20 was applied to the cells for 1 h at room temperature. The washes after each antibody were with PBS 0.2% Tween 20. The dish was sealed with a mounting medium with 4′,6-diamidino-2-phenylindole (DAPI)—aqueous (ab104139-20, Abcam, Cambridge, UK). Immunofluorescence images were acquired with Zeiss LSM780 inverted confocal microscope through a 63X objective (ZEISS International, Oberkochen, Germany).

## Figures and Tables

**Figure 1 ijms-23-06076-f001:**
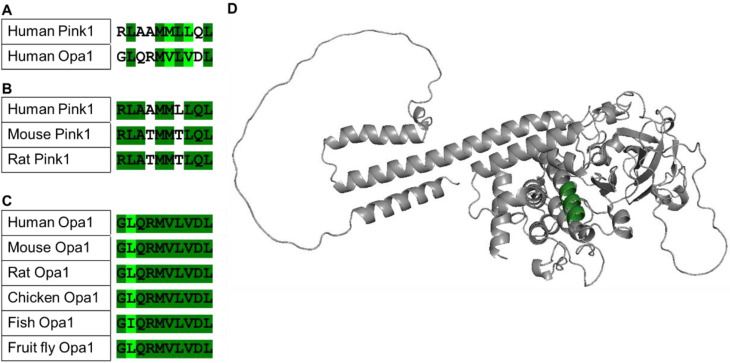
Rational design of a peptide that targets Pink1. Sequence alignment identifies a short sequence of homology between Opa1 and Pink1 (**A**), which is conserved in evolution (**B**,**C**). The Opa1 sequence (dark green) is exposed in the C-lobe region of Pink1 (AlphaFold predicted model of human Pink1: Q9BXM7) (**D**). PyMol (Schrodinger LLC) was used to generate the figure [24].

**Figure 2 ijms-23-06076-f002:**
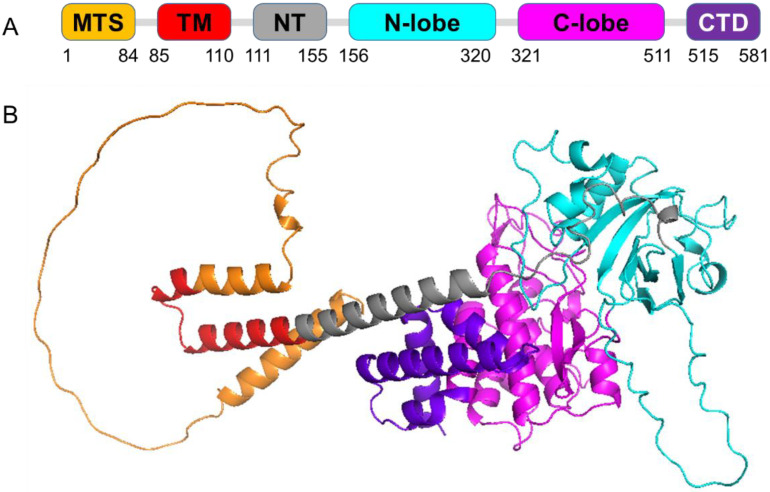
Three-dimensional (3D) predicted model of full-length human Pink1 (581 amino acid residues) (AlphaFold predicted model: Q9BXM7). Pink1 domain structure (**A**) and cartoon representation (**B**) are shown in colors that correspond to the respective domain colors. The domains are colored as follows: mitochondrial targeting sequence (MTS, orange), transmembrane region (TM, red), N-terminal regulatory region (NT, gray), N-lobe of the kinase domain (cyan), C-lobe of the kinase domain (magenta) and the C-terminal domain (CTD, purple). PyMol (Schrodinger LLC) was used to generate the figure [24].

**Figure 3 ijms-23-06076-f003:**
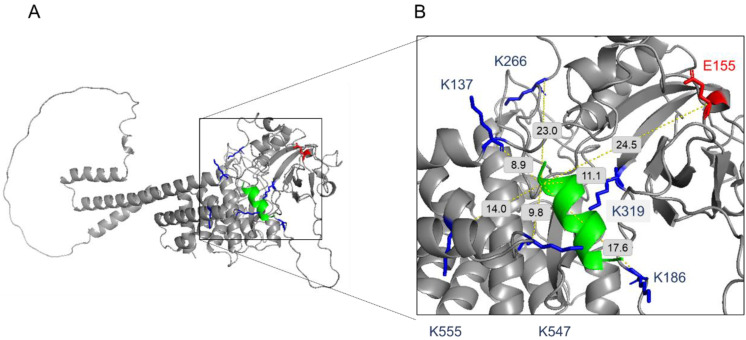
Map of the cross-link identified between the peptide CVP-198 and the hPink1 protein (AlphaFold predicted model: Q9BXM7). (**A**) Residues involved in cross-linking contacts are highlighted in blue and red on the hPink1 protein. (**B**) CVP-198 and hPink1 cross-linking contacts: CVP-198 peptide (shown in green cartoon structure) and hPink1 protein (shown in grey cartoon structure) are positioned next to each other and inter-facial cross-links, involved Pink1 lysine residues, highlighted in the blue stick structure, and involved Pink1 glutamic acid residue, highlighted in the red stick structure, are displayed (dotted yellow lines). PyMoL (Schrodinger LLC) was used to generate the figure [24].

**Figure 4 ijms-23-06076-f004:**
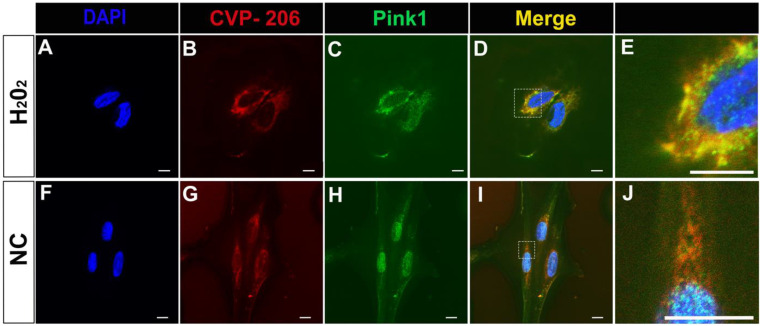
Colocalization of Pink1 and CVP-206 peptide in H9c2 cells. (**A**–**E**) Immunostaining for CVP-206 (1 µM, Red) and Pink1 (Green) after H_2_O_2_ treatment (1.6 mM for 0.5 h). (**E**) Higher magnification of D. (**F**–**J**) Immunostaining for Pink1 (Green) and CVP-206 (1 µM, Red) in H9c2 cells with no treatment, Negative control (NC). (**J**) Higher magnification of I. Nuclei are in blue (DAPI), scale bar indicates 5 µm for CVP-206 staining after H_2_O_2_, and 10 µm for negative control (NC) and MitoTracker (*n* = 3). Confocal microscopy images were taken at 63× magnification for NC and H_2_O_2_ treated cells.

**Figure 5 ijms-23-06076-f005:**
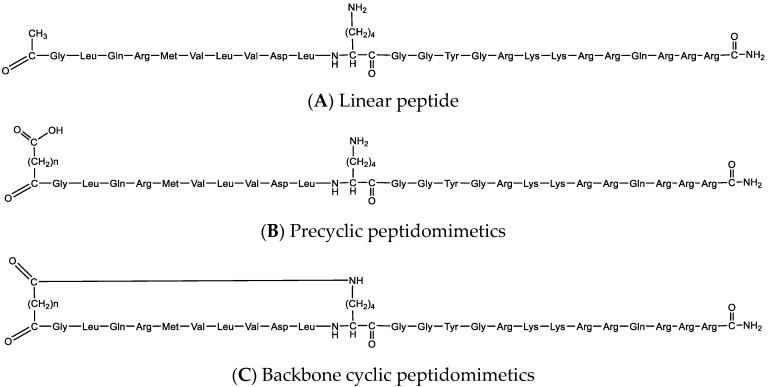
Schematic structure of the developed library. (**A**) Structure of the linear peptide, CVP-198; (**B**) structure of the precyclic peptidomimetics, CVP-201 and CVP-202; and (**C**) structure of the backbone cyclic peptidomimetics, CVP-199 and CVP-200.

**Figure 6 ijms-23-06076-f006:**
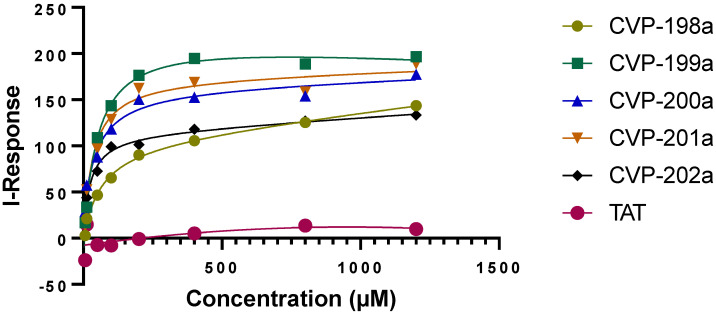
Binding curves for the interactions between linear, precyclic, and backbone cyclic peptidomimetics and the TcPink1 protein. The Y-axis corresponds to the I-Response in biosensor units (BU), and the X-axis corresponds to the different concentrations of the analyte in the experiment. The data was normalized and presented as the fraction of bound peptide. A control study was also performed in which the biomolecular interaction between TcPink1 and a control peptide, TAT, was done. It is evidenced that there is very low interaction between TcPink1 and the control peptide, TAT.

**Figure 7 ijms-23-06076-f007:**
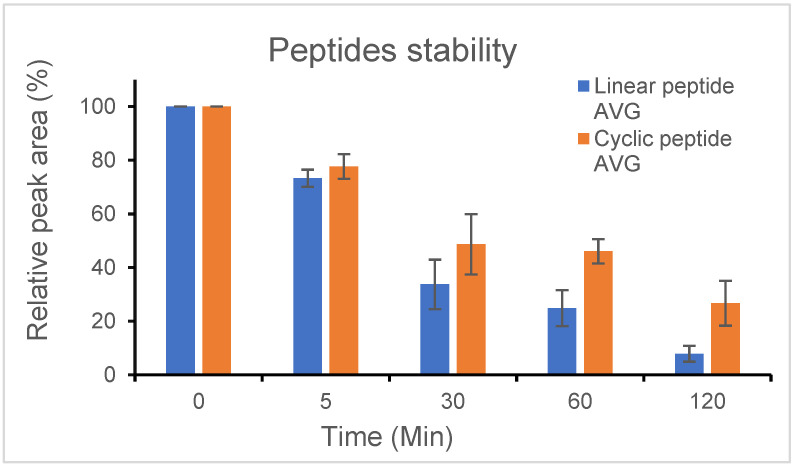
Stability studies of the linear peptide, CVP-198, and the bioactive backbone cyclic peptide, CVP-199. The stability of the peptides was measured independently under tryptic degradation at 37 °C (by trypsin degradation of the peptide) and was determined by HPLC analysis (for more experimental details, see “Materials and Methods”) (*n* = 3). The starting time point (0 min, 100%) represents the peptide at the beginning of each experiment. Linear peptide (CVP-198) at 37 °C with trypsin (blue), and cyclic peptide (CVP-199) at 37 °C with trypsin (orange), were analyzed after 0 min, 5 min, 30 min, 60 min, and 120 min. Separation was performed by HPLC in the presence of 0.1% trifluoroacetic acid and detected by absorbance at 214 nm. Peptide amounts were calculated relative to the quantities determined at time point zero.

**Figure 8 ijms-23-06076-f008:**
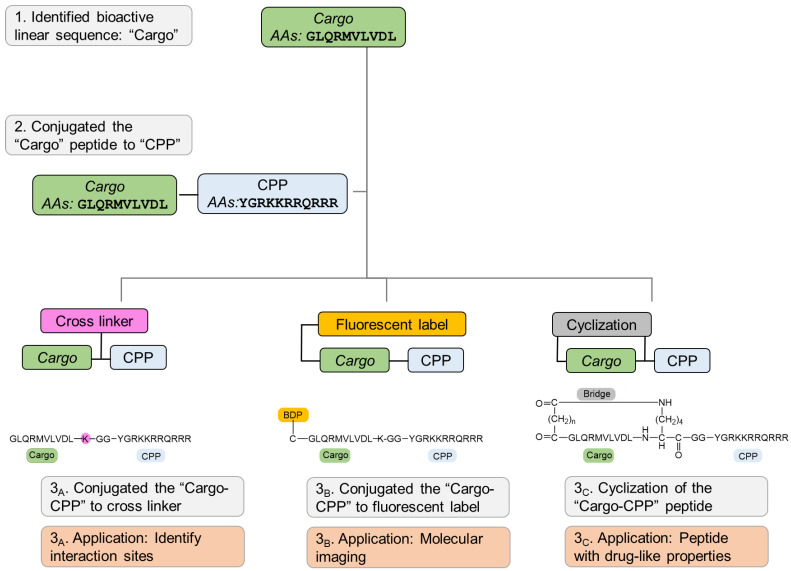
Schematic diagram of converting a bioactive linear peptide sequence into advanced research tools and potential therapeutic.

**Table 1 ijms-23-06076-t001:** Characterization of the developed library.

Peptide Name	n	Bridge Size	Ring Size	Calcd MH^+^	Observed MH^+^	Purity HPLC (%)	Notes
CVP-198	-	-	NA	2968.58	2972.54	100.00	Linear peptide
CVP-199	2	9	41	3008.61	3012.80	100.00	Cyclic peptide
CVP-200	3	10	42	3022.64	3026.07	100.00	Cyclic peptide
CVP-201	3	-	NA	3040.65	3045.26	100.00	Precyclic peptide
CVP-202	2	-	NA	3026.63	3030.11	100.00	Precyclic peptide

**Table 2 ijms-23-06076-t002:** Summary of the corresponding in vitro K_D_ values of the peptides binding to TcPink1.

Peptide Name	KD (µM)	Notes
CVP-198	130.47 ± 0.01	Linear peptide
CVP-199	58.31 ± 0.02	Cyclic peptide
CVP-200	138.72 ± 0.30	Cyclic peptide
CVP-201	125.89 ± 0.02	Precyclic peptide
CVP-202	143.09 ± 0.04	Precyclic peptide

## Data Availability

Data is contained within the article or Appendix A.

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
