# Peer review of "De Novo Development of Mitochondria-Targeted Molecular Probes Targeting Pink1"

_ijms, 2022, doi:10.3390/ijms23116076_

Round 1
Reviewer 1 Report
The manuscript “De novo development of mitochondria‐targeted molecular probes targeting Pink1” presents a description of the development of molecular probes targeting Pink1, which is suitable for publication after clarifying a few things.
Indeed it is interesting to engineer a molecular probe to target mitochondria protein to elucidate its role in mitophagy and cell fate activities. But it is kind of disappointed to see that only “weak” binding is observed for the designed peptide with an unknown specificity (Kd around 50 uM without any data on the specificity). Of course it is not, by any means, that this should not be published. However, it would be beneficial that authors try to provide stronger evidence that this molecular probe works good enough as a molecular probe or potentially as a therapeutic agent. Some points and experiments I suggest if the authors are willing to take.
- The introduction part is kind of thin and weak. At least general events in the PINK1 pathway should be described to give a brief introduction on how PINK1 works and which proteins and factors are involved in its function. Also, current molecular probes for PINK1 should be described. What are their problems and why a new molecular probe is needed? Although there may not be many but it is still valuable to discuss about this to give a better rationale for this paper. Also why authors choose specific cell penetrating peptides and why to engineer these peptides need
a detailed description.
- The docking part of the peptide is not clearly described. The authors need to specify the type of docking they have carried out (rigid, flexible, induced fit?). What are the results? Simply stating that “the peptide binds with the C‐lobe, the CTD together with random coils of the N‐lobe” is not clear enough to convince readers that this peptide is a real binder and how strong the binding is? Is there any other site the peptide will bind to?
- Figure 3 basically do not contain much information. No detailed interaction or binding pockets are shown. I would suggest putting it in the supplement.
- Line 128-130 you do not need to describe how important CLMS is. This is more like an introduction.
- Although this is no doubt conveniently explained in figure 6 and table 1, I think the manuscript would benefit from adding some sequence comparison information showing how the different peptides tested differ from each other.
- The fluorescent probe part I think it will be more convincing if the authors contain some control proteins for comparison. Proteins shown to interact with PINK1 should be included such as PARKIN, Opa1 and some PINK1 trafficking proteins to show this probe is specific to PINK1.
- Figure 6 is poorly drawn. Some bonds and atoms are twisted in the figure. I suggest using Chemdraw to redraw the figure.
- The cyclization part, it would be valuable the authors explain how the specific bridge size is chosen. Why just n=2 or 3 is synthesized? Do the authors have any preliminary data to support the choice?
- Table 2 is mostly duplicate as Table 1 except the Kd results, which can be merged. Purity part in Table 1 is kind of less useful since all peptides are 100% pure. Also the calculated MS and observed MS seems not a big issue that authors want to discuss in the main context. I suggest them to be moved to the supplement.
- The “Binding of linear and backbone cyclic peptides” part, as the binding for the proposed peptides are only mediocre. It will be more convincing that authors also test these peptides on PINK1 related proteins to prove they are specifically bond to PINK1.
- The stability part, I am wondering if the authors do a triplicate on all peptides as there is no standard deviation on Figure 8. If not, I would suggest including that. Also it would be interesting authors make an explanation on why linear peptide seems gain increased stability after 120 min.
Author Response
Attached is our revised manuscript submitted to be considered for publication in the International Journal of Molecular Sciences (IJMS). We would like to thank the Reviewers and Editors for taking the time to provide us with a thorough review and positive feedback. We did our utmost to address the below concerns of the reviewers as adequately, and in as detailed a manner, as possible. We believe this manuscript is improved and submit it for publication in the International Journal of Molecular Sciences. Below is a point-by-point response (in blue) to the reviewers’ comments (brought verbatim in black).
Thank you for your consideration.

Reviewer 2 Report
Authors should do co-immunoprecipitation or pull down experiment to validate the interaction between Pink1 and Opa1.
Author Response

(The authors gave the same response as above.)

Reviewer 3 Report
In the present manuscript “De novo development of mitochondria-targeted molecular probes targeting Pink1”, Shulamit Fluss Ben-Uliel and colleagues rationally designed a linear peptide that targets Pink1 and, using straightforward chemistry, they developed molecular probes with drug-like properties to further characterize Pink1. The authors concluded that this represents a general approach using straightforward chemistry that can be tailored for various applications by numerous laboratories. Overall, I think that the manuscript is well-written (within the scope of this journal), well-structured and the data are of potential relevance on a current topic of interest.
I have little suggestions to improve the quality of paper.
1) In light of the results obtained in the present study and from a translational point of view, please to deeply discuss on the possible impact of this innovative approach in the field of “nutrition, functional foods, bioactives and nutraceuticals” (specific topic of this section).
2) Indeed, the authors could update the Figure 9 considering the previous observation; in this way, I feel that the readers can better understand the approach studied in the present paper and the possible application for basic research as well as lead molecules useful in drug discovery.
Author Response

(The authors gave the same response as above.)
